# Management of Advanced Pancreatic Cancer through Stromal Depletion and Immune Modulation

**DOI:** 10.3390/medicina58091298

**Published:** 2022-09-17

**Authors:** Tiantong Liu, Sihang Cheng, Qiang Xu, Zhiwei Wang

**Affiliations:** 1Department of General Surgery, Peking Union Medical College Hospital, Chinese Academy of Medical Sciences, Beijing 100006, China; 2Department of Radiology, Peking Union Medical College Hospital, Chinese Academy of Medical Sciences, Beijing 100006, China

**Keywords:** advanced pancreatic cancer, tumor microenvironment, stromal depletion, immune modulation

## Abstract

Pancreatic cancer is one of the leading causes of cancer-related deaths worldwide. Unfortunately, therapeutic gains in the treatment of other cancers have not successfully translated to pancreatic cancer treatments. Management of pancreatic cancer is difficult due to the lack of effective therapies and the rapid development of drug resistance. The cytotoxic agent gemcitabine has historically been the first-line treatment, but combinations of other immunomodulating and stroma-depleting drugs are currently undergoing clinical testing. Moreover, the treatment of pancreatic cancer is complicated by its heterogeneity: analysis of genomic alterations and expression patterns has led to the definition of multiple subtypes, but their usefulness in the clinical setting is limited by inter-tumoral and inter-personal variability. In addition, various cell types in the tumor microenvironment exert immunosuppressive effects that worsen prognosis. In this review, we discuss current perceptions of molecular features and the tumor microenvironment in pancreatic cancer, and we summarize emerging drug options that can complement traditional chemotherapies.

## 1. Introduction

Pancreatic cancer accounts for 6–8% of annual cancer-related mortalities [1]. The mortality and incidence rates are nearly the same due to the heterogeneous nature of pancreatic cancer, which equates to a poor prognosis in most patients [2]. Despite the robust growth in basic research on pancreatic cancer and improvements in surgical techniques, the 5-year survival rate for patients remains relatively low at approximately 8–9 per 100,000, which is lower than the survival rates for most other cancers [3].

The obstacles in managing pancreatic cancer are multidimensional. Pancreatic cancer is a heterogeneous array of pathological and molecular conditions rather than a singular disease, which leads different patients to respond differently to the same treatment [4,5]. The lack of accurate, cost-effective screening strategies makes pancreatic cancer more difficult to diagnose early than breast, colorectal, and lung cancers [6]. Most patients with early-stage, localized pancreatic cancer present no symptoms or only mild, non-specific symptoms such as fatigue, epigastric discomfort, and loss of appetite. Progressive abdominal pain, weight loss, and jaundice are usually the symptoms that finally raise the alarm for patients and healthcare providers, but by this point, the disease has typically reached an advanced stage that is difficult to treat. By the time of diagnosis, only 20–30% of patients are eligible for surgical resection, and up to 50% may already be at an unresectable metastatic stage [7]. Even for the small subset of patients who can undergo surgical resection, the 5-year survival rate remains lower than 20% [8]. Distal metastasis occurs early during pancreatic cancer progression [9], and metastatic pancreatic cancer is associated with a mortality rate of 95% and median overall survival shorter than one year [9].

The traditional first-line treatment for patients with unresectable pancreatic cancer is the classical chemotherapy gemcitabine. However, many patients develop resistance to this drug within weeks of treatment initiation [10]. Whereas multiple forms of chemotherapy or targeted therapies are available for other cancers, treatment options for pancreatic cancer are severely limited.

Successful cancer management is based on a comprehensive molecular and cytological understanding of tumor cells. Technological innovation in biomedical research has profoundly enriched our knowledge about pancreatic tumorigenesis and progression over the last decade [6,11]. However, a series of promising drug targets have been discovered and may fill gaps in the treatment of advanced pancreatic cancer and enable individualized cancer therapy.

This review briefly summarizes the current treatment recommendations for pancreatic cancer, describes critical advances in subtyping pancreatic cancer and understanding the tumor microenvironment, and presents clinical trials focusing on relevant drug targets.

## 2. History and Current Recommendations of Advanced Pancreatic Cancer Management

Cytotoxic therapy has been the mainstay of advanced cancer management because it extends patient survival time, albeit only for a limited period of time. Gemcitabine is a hydrophilic nucleoside analog that can be transported into cells to disrupt normal DNA synthesis and inhibit the proliferation of tumor cells. Since the early 1990s, gemcitabine has been the standard of care for advanced pancreatic cancer because it showed better outcomes than fluorouracil [12]. Efforts to improve the outcomes of gemcitabine by combining it with cytotoxic agents or targeted drugs have shown some success [13,14,15,16,17,18]. Erlotinib, an inhibitor of the epidermal growth factor receptor widely used to treat unresectable non-small cell lung cancer, can improve the overall survival of patients with advanced pancreatic cancer by 0.33 months, but at the expense of greater adverse effects [19]. Nab-paclitaxel increases gemcitabine concentrations in the blood and significantly improved response rate and median survival in a phase III trial [20,21]. The US Food and Drug Administration approved this combination therapy in 2013 for advanced pancreatic adenocarcinoma. In 2017, gemcitabine and capecitabine proved to be an effective combination of chemotherapy among pancreatic cancer patients who have undergone pancreatic tumor resection [22]. The combination of leucovorin, fluorouracil, irinotecan, and oxaliplatin, a regimen known as “FOLFIRINOX”, prolonged median overall survival from 6.8 months to 11.1 months [23]. As a result, several countries incorporated FOLFIRINOX into their guidelines for pancreatic cancer management, yet its severe toxicity makes it suitable only for certain patients.

Current pancreatic cancer treatment takes a multi-pronged approach [24]. Pancreatic cancer can be divided into four basic categories based on anatomical resectability: resectable, borderline resectable, unresectable (locally advanced), and metastatic. The classification largely depends on high-resolution computed tomography and surgeon experience [25]. Neoadjuvant chemotherapies can improve the resectability of borderline resectable pancreatic cancer [26]. For unresectable pancreatic cancer, the recommended first-line treatments are combinations of gemcitabine with FOLFIRINOX, nab-paclitaxel, or erlotinib. If gemcitabine-based treatment fails upon cancer metastasis, recommended second-line therapies include: the combination of fluorouracil, folinic acid, and oxaliplatin (OFF); nanoliposomal irinotecan, with or without fluorouracil and folinic acid; and the combination of folinic acid, oxaliplatin, and fluorouracil (FOLFOX) [27,28,29].

## 3. Tumorigenesis and Tumor Microenvironment in Pancreatic Cancer

### 3.1. Hallmarks of Pancreatic Cancer

Cancer cells share typical hallmarks that distinguish them from normal cells, and cancers of different origins demonstrate unique characteristics that are potentially targetable [30]. Pancreatic cancer cells carry specific genomic alterations critical to tumorigenesis and progression [31]. Classically, pancreatic cancer tumorigenesis has been considered the result of a sequential accumulation of molecular perturbations in resident pancreatic acinous cells that lead to intraepithelial neoplasia and, ultimately, cancer [32,33]. These disturbances include: mutations in KRAS, TP53, CDK4/6, and BRCA1/2; deletion of BRAF; and gene rearrangements of NTRK, ALK, and NRG1 [32,34]. However, clinical efforts to target various genomic alterations for treatment have so far proved to be disappointing [35].

Metabolic reprogramming is another remarkable feature of pancreatic cancer cells [30]. The hypoxic intra-tumoral environment leads tumor cells to shift their metabolism from oxidative phosphorylation toward oxygen-independent glycolysis, known as the “Warburg effect” [36,37], to help tumor cells survive the scarcity of nutrients and oxygen in their microenvironment. As part of this metabolic shift, redundant glycolytic intermediates enter various metabolic bypass pathways [38,39], and glutamine transportation is hyperactivated [40]. Autophagy and macropinocytosis are hyperactivated to scavenge nutrients [41,42,43,44]. Novel therapeutic opportunities exist in inhibiting cancer cell mitochondrial bioenergetics or scavenging pathways, as reviewed previously [45].

### 3.2. The Inter-Tumoral Microenvironment of Pancreatic Cancer

In the pancreatic tumor microenvironment, the extracellular matrix (ECM), immune and stromal cells, and secretory signals interact to promote cancer cell invasion, dissemination, and distal colonization [42,46]. Cancer-associated fibroblasts (CAFs) in the tumor microenvironment create a desmoplastic extracellular matrix that helps drive pancreatic cancer cell progression and drug resistance by secreting various fibrotic substances, such as collagen and hyaluronic acid [47]. CAFs also secrete various amino acids and lipids that support cancer cell survival [48,49,50]. There are at least two types of CAFs within the tumor microenvironment. Myofibroblasts are characterized by α-smooth muscle actin (α-SMA), and inflammatory CAFs lose myofibroblastic features and secrete various inflammatory cytokines [51,52].

Immune cells in the PTME play a critical role during cancer progression. Compared to other types of cancer, most pancreatic cancers, if not all, are immunologically cold, which is represented by the limited infiltration of effector T cells and low expression of PD-L1, thereby eliciting low responses to immune checkpoint inhibitor therapy [53]. The restrained proliferation and activation of cytotoxic T cells are closely related to the composition of immune mi, mainly comprising macrophages, myeloid-derived suppressor cells (MDSC), fibroblasts, neutrophils, and T regulatory cells [54,55].

Tumor-associated macrophages (TAMs) are key components of cancer stroma inflammation, presenting a spectrum of functional states with two opposing extremes [42]. While polarized M1 macrophages are pro-inflammatory and anti-neoplastic, M2 macrophages have the opposed phenotype, which promotes tumor growth and predominates during cancer development [42]. Multiple lines of evidence have shown that macrophage depletion efficiently impairs angiogenesis and reduces metastasis formation [56]. Inhibition of TAMs also alleviates gemcitabine resistance by inactivating CDA, an enzyme critically involved in the inactivation of gemcitabine [57].

Myeloid-derived suppressor cells (MDSCs) are strongly increased in both the circulation and the TME of PDAC patients, and a positive correlation exists between the cell count and tumor staging [58]. Granulocyte-macrophage colony-stimulating factor (GM-CSF) and IL-1β secreted by tumor cells are critical to the development of MDSCs [59]. MDSCs increase the expression of tumor PD-L1 expression via the activation of EGFR/MAPK signaling pathway and result in CD8+ T cell exhaustion and T regulatory cell expansion [58,60], thus abrogation of tumor-derived GM-CSF secretion prohibits the MDSCs infiltration and blocks tumor development by increasing the number of cytotoxic T cells [61].

The role of neutrophils educated by cancer cells, or tumor-associated neutrophils (TAN), has received less attention in the PTME. Evidence showed that TANs contribute to pancreatic cancer cell spreading by producing multiple enzymes, namely matrix metalloproteinase (MMP) 8 and MMP9, Cathepsin-B, and proteinase-3, which accelerate the degradation of the ECM, break down the barrier, and promote cancer invasion and metastasis [62]. Neutrophil extracellular traps (NETs) are responsible for the exclusion of cytotoxic CD8+ T cells from tumors and may have a role in driving pancreatic cancer metastasis [63,64]. In preclinical models, lorlatinib, a TANs suppressor, attenuates pancreatic cancer growth and improves the efficacy of immune checkpoint inhibitors [65]. Similarly, the inhibition of neutrophil activator IL17 increased immune checkpoint blockade (PD-1, CTLA4) sensitivity [63].

The ECM in the pancreatic tumor microenvironment also appears to contribute to immunosuppression [66]: the thick, fibrotic ECM inhibits tumor vascularization and immune cell infiltration, as well as the delivery of drugs into tumors [67]. Thus, a suggested strategy is to bring back the normalization in the tumor immune ecosystem and make the immune backbone of the tumor hot or immunogenic so that it would be more responsive to therapy [53,68].

Although some studies have shown that an immunologic subtype of pancreatic cancer does exist and generally has a better prognosis than the unselected patient population [69,70], the treatment of immunosuppressive PDAC remains a major obstacle. The targeting of various contributors of the immune microenvironment in combination with immune checkpoint inhibitors is promising for synergistic antitumoral effects in preclinical models of pancreatic cancer and some ongoing clinical trials.

## 4. Molecular and Cellular Subtyping of Pancreatic Cancer

Morphologically and pathologically indistinguishable pancreatic tumors can have quite different genomic features and respond differently to antitumor therapy. Thus, a molecular classification system is required to guide clinical management adequately [71].

Molecular subtyping is a necessary stepping stone toward individualized anticancer therapy. Early attempts to classify pancreatic cancers relied on single genetic markers, including point mutations, structural variations, and protein markers [34,72,73,74,75,76,77]. Though most classification schemes have failed to enter the clinic, some markers, such as KRAS, TP53, and ERBB2, have been validated as prognostic markers and surgical indications [78]. In 2011, microarray data were used to define three subtypes of pancreatic ductal adenocarcinoma (PDAC): “classical”, “quasimesenchymal”, and “exocrine-like”. Each subtype shows distinguishable histopathological features that may indicate prognosis and responsiveness to gemcitabine and erlotinib [79]. More recently, an “immunogenic” subtype has been described, and is characterized by strong immune infiltration of the tumor microenvironment [33,79].

Pancreatic cancer has been subtyped as “classical” or “basal-like” based on its cellular composition [80,81]. The classical subtype arises from an endodermal-like stromal lineage and shows activation of the transcription factor GATA6 and KRAS expression. In contrast, basal-like cells are associated with altered chromatin modification and a worse prognosis. Classical PDAC appears to respond better to first-line pancreatic cancer chemotherapy than basal-like PDAC [82]. More recent work has identified additional cell-based subtypes, including “desmoplastic”, “immune classical”, and “stroma-activated” [71].

Multi-omics approaches may be key to elucidating on the complex nature of pancreatic cancer. Indeed, integrating genomic and transcriptomic information revealed that the classical and basal-like subtypes still contain substantial heterogeneity, which should be studied further [83]. For example, the basal-like subtype may contain a “squamous” subpopulation associated with altered chromatin modification and faster disease progression [84,85]. Comprehensive genomic and transcriptomic profiling may help pave the way for personalized treatments against pancreatic cancer.

## 5. TME Normalization for the Management of Advanced Pancreatic Cancer

The highly desmoplastic tumor stroma and presence of desmoplasia-inducing stromal cells are major contributors to the lack of efficacy of, and resistance to, various anticancer drugs [86]. Hedgehog (Hh) signaling is abnormally activated in pancreatic intraepithelial neoplasia and PDAC, and it is critical for pancreatic cancer stroma formation and stabilization [87,88]. Blocking Hh in mouse models increased the tumor vascularization and intra-tumoral concentration of gemcitabine [89]. However, adding a Hh inhibitor to either gemcitabine alone or FOLFIRINOX failed to improve outcomes for advanced pancreatic cancer patients [90,91,92]. This result was understandable because the Hh pathway involves many other biological and pathological processes. An ongoing clinical trial investigates the safety and efficacy of inhibiting focal adhesion kinase (Table 1), a crucial regulator of the fibrotic and immunosuppressive pancreatic tumor microenvironment [93]. However, inhibiting a specific signal transduction pathway on its own may prove ineffective: for example, inhibiting focal adhesion kinase appears to promote drug resistance [94]. Drug targets with more specific antitumor effects are warranted.

The cell–cell communication between the tumor cell and the non-tumor stromal constituents promotes the TME to shift towards a ‘hungry’ state that largely promotes tumor growth, invasion, and metastasis. Various stromal cells and immune cells present with cancer-associated phenotypes [42], and induced neovascularization is a shared hallmark of the TME of almost all solid tumors in response to the increased need for oxygen and nutrients [95,96]. These alterations form the fertilized “soil” that facilitates tumor growth. Rather than directly developing novel therapies that present a higher potency to kill tumor cells, “stroma normalization strategies” aim at reversing the TME state to a relatively normal one from different angles by increasing the function of vessels or modulating the phenotype of various populations of stromal cells in order to enhance the efficacy of well-established chemo-, radio-, and immunotherapies. CAF reprogramming, vascular reconstruction, and immune modulation are three promising ways that TME normalization strategies can manage solid tumors, which will be elaborated on in the following sections.

### 5.1. CAF Reprogramming and ECM Depletion

Targeting CAFs may be another way to control tumor growth [50], such as by depleting them, reprogramming them to adopt an anti-tumorigenic or quiescent state, or inhibiting their communication with the tumor stroma [97,98,99,100]. Inhibition of molecular markers of CAFs, such as α-SMA and fibroblast activation protein (FAP), can disrupt desmoplasia in the pancreatic tumor microenvironment and block tumor growth [101]. However, targeting CAFs or the fibrotic tumor microenvironment is a difficult issue. For unknown reasons, disruption of the tumor microenvironment’s desmoplastic nature can lead to anticancer treatment failure [94]. Deleting α-SMA-expressing myofibroblasts in pre-cancerous or PDAC tumors in mice exacerbated hypoxia and suppressed immune cell infiltration, which reduced survival [102]. The multiple roles of CAFs make it difficult to predict the effects of disrupting them [4]; an alternative approach may be to target specific signals secreted by CAFs. For example, pharmacologic depletion of CXCR4 and immunotherapy inhibited cancer progression in a mouse model of PDAC [103]. A phase II trial suggested that CXCR4 inhibition combined with pembrolizumab can reduce populations of immunosuppressive cells in the tumor microenvironment of patients with chemotherapy-resistant advanced pancreatic cancer [104].

The therapeutic rationale for disrupting constituents of the ECM is to break down the mechanical barrier forged by dense desmoplasia. During cell differentiation, fibroblasts secrete major components of the ECM Hyaluronan, collagen, and fibronectin as physical barriers to impede the transport and diffusion of antitumor drugs [67]. Therefore, targeting these fibrosis-related components should increase such drugs’ local concentration and efficacy. Hyaluronidase degrades hyaluronan in the ECM and decreases interstitial fluid pressure [105]. Recombinant hyaluronidase has been tested as a complementary therapy to enhance combination chemotherapy [106,107], but preclinical and clinical studies have given conflicting results. For example, the hyaluronidase PEGPH20 improved the benefits of chemotherapy in a phase II trial, yet it increased rates of drug-related adverse effects in a subsequent phase III trial [108,109,110,111]. Therefore, targeting components in the ECM may be insufficient to overcome drug resistance. Efforts are underway to explore the simultaneous pharmaceutical inhibition of multiple distinct pathways or biological processes, particularly immunomodulatory pathways (Figure 1).

### 5.2. Anti-Angiogenic Therapy and Vascular Normalization

Although angiogenesis is extensively induced in solid tumors, the mismatch between oxygen and nutrient supply and cancer cell division largely creates a hypoxic intra-tumoral environment [95]. Hyperpermeability and branch tortuosity of the newly formed vessels, as well as the increased physical compression by uncontrolled expansion of tumor cell mass, further compromise the blood perfusion of the tumor, which limits the access of various immune cells [53]. Therefore, strategies that alter the intra-tumor vasculature have long been considered for their antitumor potential. The earliest therapy targeting tumor vasculature was the anti-vascular endothelial growth factor (anti-VEGF) agents, such as bevacizumab, which has been shown to improve the efficacy of traditional chemotherapy or immunotherapy for some solid tumors [112,113]. Nevertheless, this success cannot be replicated in pancreatic cancer because multiple phase III trials with anti-VEGF agents have failed to increase the overall survival of advanced pancreatic cancer patients compared to chemotherapy alone [114]. In several phase III trials, other regimens of the same category, such as sorafenib, axitinib, and ZIV-aflibercept in combination with gemcitabine, also failed to improve the outcome [115]. Though the mechanism remains incompletely understood, the simple hypovascularized nature of pancreatic ductal adenocarcinoma is not enough to account for the failure [69]. Using a PDAC mouse model, Aguilera et al. found that chronic treatment with bevacizumab-induced intra-tumoral hypoxia accelerated collagen deposition and increased the overall tumor burden [116].

Instead of a simple blockade of vasculature genesis, tumor vasculature normalization is the concept of eliminating excess endothelial cells, therefore pruning immature and unproductive vasculature to provide easier access for the anticancer drug delivery to cancer cells. Furthermore, the normalization can increase pericyte coverage, fortify the immature vessels, and improve tumor perfusion. In preclinical animal models, Semaphorin 3A (SEMA3A) was secreted by endothelial cells and acted as a vasculature-normalizing regulator by negatively impacting integrin expression [117]. Expression of SEMA3A gradually decreases during the transition of premalignant lesions to the tumor, consistent with the development of a dysfunctional abnormal tumor vasculature system during this course [117]. Restored expression of SEMA3A resulted in the normalization of the tumor vasculature, impairment of metastatic disease progression, and sensitization of immunotherapy [118,119]. Moving forward, anti-angiogenic therapy in PDAC should consider proper experimental models that recapitulate the angiogenic feature of the real tumor and possibly classify the pancreatic cancers in terms of vasculature extent to improve the benefit of therapy.

### 5.3. Immune Checkpoint Blockade (ICB)

Adoptive cell transfer and immune checkpoint blockade (ICB) are two powerful treatment paradigms that have achieved groundbreaking success in multiple cancers, such as melanoma and leukemia [42]. However, both have been largely unsuccessful against pancreatic cancer [42]. A single classical ICB has an average response rate of only 5% in treating advanced pancreatic cancer [120]. Nevertheless, preclinical and clinical trials continue in PDAC animal models and patients to target tumor antigens or components of the tumor microenvironment [121,122,123]. Routine evaluation of tumor mutational burden has been recommended to identify patients that may potentially benefit from ICB [124]. For example, CD8 and T cell receptor clonality expression have been correlated with responses to ICB targeting PD-1/PD-L1 [125]. Innate lymphoid cells are involved in activating tissue-specific tumor immunity in PDAC [126]. “Switchable” adoptive cell transfer, which exploits a recombinant peptide that bridges tumor antigen and the Fab domain of the T-cell receptor (TCR) to control CAR-T cell response, can potentially achieve tunable antitumor and avoid off-tumor effects in preclinical PDAC models [127]. This strategy potentially improves the safety of CAR-T therapy in solid tumors while preserving antitumor responses to the largest extent.

### 5.4. Pancreatic Tumor Vaccine

“Cancer vaccine” strategies aim to increase the efficacy of existing ICBs by using immunostimulatory agents to sensitize the host immune system against tumors. One potential vaccine, GVAX pancreas, has been formulated with allogeneic pancreatic tumor cells that secrete granulocyte-macrophage colony-stimulating factor (GM-CSF) to induce T cell infiltration of the pancreatic tumor. In contrast, the vaccine CRS-207 contains the live attenuated pathogen Listeria monocytogenes, which stimulate an immune response against mesothelin, a tumor-associated antigen overproduced by the pancreatic tumor [128,129]. One trial failed to demonstrate the clinical benefit of the combination of GVAX pancreas, cyclophosphamide, and CRS-207 against advanced pancreatic cancer [130,131]. The whole-cell immunomodulator HyperAcute-Pancreas algenpantucel-L (HAPa) comprises allogenic pancreatic cancer cells that elicit antibody-dependent cell-mediated cytotoxicity [132]. However, it failed to improve the survival of patients with borderline resectable or locally advanced, unresectable PDAC who received chemotherapy alone or with radiation in a phase III trial [132]. Conversely, adding a tumor vaccine targeting VEGFR1/2 or personalized neoantigen peptides improved the efficacy of gemcitabine [133,134]. Although vaccines or immune-enhancing approaches on their own may be ineffective, their combinations should be explored further.

### 5.5. Cytokine-Based Therapy

Another immune-modulating strategy is the regulation of immunosuppressive cells in the pancreatic tumor microenvironment, such as regulatory T cells, cancer-associated macrophages, and myeloid-derived suppressor cells [58,135,136,137,138]. Activation of classical tumor necrosis factor receptor family member CD40 enhances antigen presentation by dendritic cells and activates cytotoxic T cells in animal models [139]. CD40 agonists can also polarize macrophages in the tumor microenvironment to adopt a pro-inflammatory state that can kill tumors in preclinical studies [140,141]. Combining CD40 priming with other immune modulation may enhance antitumor immune responses [142] and potentiate ICB and chemotherapy [143,144]. However, adding the monoclonal antibody sotigalimab or nivolumab, which acts as a CD40 agonist, to gemcitabine and nab-paclitaxel failed to provide satisfactory efficacy in a phase II trial [145,146]. This may reflect that gemcitabine antagonizes the ability of CD40- and ICB-based therapy to restrict tumor progression in animal models [147].

Cytokines and chemokines regulate the differentiation and maturation of various immune cells. In mouse models, inhibiting CCR2 and CSF-1R can antagonize the ability of tumor-associated macrophages to suppress T cell responses [148,149]. However, whether combining CCR2 inhibitors with nab-paclitaxel plus gemcitabine is effective in patients remains unclear [150]. In fact, the combination of the CSF-1R inhibitor cabiralizumab, the PD-1 inhibitor nivolumab, and gemcitabine failed to improve the progression-free survival of advanced pancreatic cancer patients compared to gemcitabine alone. This may reflect that CAFs induce tumor infiltration by CXCR2-expressing MDSCs [151], which argues for multi-pronged therapy. IL-10 activates cytotoxic T cells and stimulates their proliferation [105], yet recombinant human IL-10 has not been proven effective against advanced chemotherapy-resistant PDAC [152].

### 5.6. STING DNA-Sensing Pathway Agonists

Stimulator of Interferon Genes (STING) is a transmembrane endoplasmic reticulum protein normally activated by cytosolic DNA, which binds to cGAMP and activates transcriptional gene cascades and ultimately results in type I interferon (IFN) production. It is a key regulator for generating cytotoxic T cells [153]. Activation of the STING pathway in antigen-presenting cells (APCs) is essential for checkpoint blockade and anti-PD1 therapies [154]. In preclinical models, systemic or intra-tumoral administration of STING agonists is powerful in reversing immune suppression of the tumor and improving cytotoxic T-cell infiltration [154,155,156]. STING-dependent vaccines can inhibit tumor growth and improve long-term antitumor memory [157]. As one of the earliest investigated STING agonists, 5,6-dimethylxanthenone-4-acetic acid (DMXAA) can modulate the immune system and result in anticancer responses in mice [154]. Similarly, cytosolic cyclic dinucleotides (CDNs) enhance type I interferons’ production by activating TBK1/IRF3 and NF-κB pathways [158]. More evidence is emerging that STING activation is an effective antitumor strategy that can be applied in various forms.

## 6. Challenges and Future Perspectives

Pancreatic cancer is one of the deadliest malignancies worldwide due to limited therapeutic options. Treating and managing it effectively will require a multidisciplinary effort, including improvements in basic research, disease screening and diagnosis, drug development, surgical techniques, and, most importantly, evidence-based clinical decision-making. Progress in elucidating intra- and inter-tumoral heterogeneity has identified several promising therapeutic strategies, such as stromal depletion, immune modulation, and pathway targeting.

Effective therapies will most likely depend on combining two or more of these therapies at once. For example, treatments targeting only stromal depletion or immune modulation have generally proven disappointing in clinical trials. In part, this reflects the dynamic crosstalk between the stroma and immune cells in the pancreatic microenvironment, such that alterations in one can lead to dramatic changes in the other. Combining stroma-targeting and immune-targeting treatments may be more effective: for example, cancer vaccine-induced T cell infiltration could be enhanced by the hyaluronidase PEGPH20 in murine models [159].

Several pancreatic cancer treatments that showed promise in preclinical research proved ineffective or harmful in patients. This reflects the difficulty in creating preclinical in vitro and in vivo models that emulate the clinical characteristics of pancreatic cancer. For example, the earliest pancreatic cancer studies used 3-methylcholanthrene as a carcinogen to induce pancreatic tumor formation in wild-type C57BL/6 mice, and the widely used Panc02 cell line was derived from these animals [160]. This in vitro pancreatic cancer cell model was flawed due to the absence of hallmark gene mutations that we now know are important in pancreatic tumorigenesis [31]. This limitation was overcome by genetically engineering the KPC mouse model to include the mutations and more closely mimic human cancer progression [161]. Preclinical studies of stroma-depleting therapies have also failed to model the tumor stroma accurately. The stroma exerts complex influences on pancreatic tumors, which in different contexts can suppress tumor growth or promote metastasis [162,163]. Desmoplasia in the tumor microenvironment can vary substantially between patients, between tumors in the same patient, and even within the same tumor [4,80,164]. The same stromal depletion approach may lead to opposite outcomes in different contexts, highlighting the need for individualized assessment of patient suitability for stroma-depleting therapies.

## 7. Conclusions

In conclusion, creating therapeutic possibilities for patients with advanced pancreatic cancer will require deepening our biological and molecular understanding of the disease while improving the preclinical development and clinical translation of promising treatments.

## Figures and Tables

**Figure 1 medicina-58-01298-f001:**
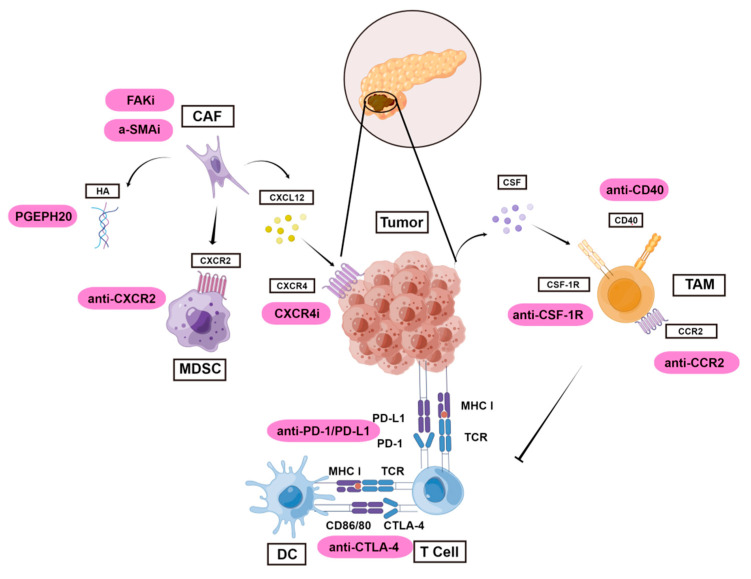
Landscape of immunomodulatory and stroma-depleting strategies to treat pancreatic cancer. Various immune cells and stromal cells constitute the pancreatic tumor microenvironment and can provide druggable targets. Promising multi-modal therapy usually involves two mechanistically independent immunomodulatory strategies or one immunomodulatory therapy combined with stroma-depleting therapy. Enhancing antigen presentation, depleting or inhibiting immunosuppressive components, or activating cytotoxic T cells are immunomodulatory strategies that can reinforce the antitumor efficacy of other therapies. APC, antigen-presenting cell; α-SMA, α-smooth muscle actin; CAF, cancer-associated fibroblast; CCR, chemokine receptor; CSF, colony-stimulating factor; CSF-1R, colony-stimulating factor-1 receptor; CTLA-4, Cytotoxic T-lymphocyte antigen 4; CXCR, C-X-C chemokine receptor; DC, dendritic cell; FAK, focal adhesion kinase; FAP, fibroblast activation protein; HA, hyaluronic acid; MDSC, myeloid-derived suppressor cell; MHC, major histocompability complex; TAM, tumor-associated macrophage.

**Table 1 medicina-58-01298-t001:** Ongoing clinical trials involving potential therapies against pancreatic cancer that target the tumor microenvironment.

Trial Identifier	Drug	Cancer Stage	Mechanism	Phase	Starting Year	Status *	Notes
**Therapeutic strategy: Stromal depletion**							
NCT03941093	Pamrevlumab	Locally advanced PDAC	Antifibrotic	III	2019	Active	With or without chemo
NCT03634332	PEGPH20+Pembrolizumab	Advanced PDAC	hyaluronidase+PD-1 inhibitor	II	2018	Recruiting	No control group
**Therapeutic strategy: Immune modulation**							
NCT02345408	CCX872	Unresectable PDAC	CCR2 inhibitor	Ib	2015	Active	With or without chemo
NCT02767557	Tocilizumab	Advanced PDAC	IL-6 inhibitor	II	2016	Active	With chemo (Gem+Nab-P) compared to chemo alone
NCT03184870	BMS-813160	Advanced PCa and CRC	CCR2/5 inhibitor	I/II	2017	Active	With or without chemo or nivolumab
NCT03941093	ABBV-927	Advanced PDAC	CD40 inhibitor	II	2019	Active	With or without chemo
NCT03336216	Cabiralizumab+Nivolumab	Advanced PDAC	CSF-1R inhibitor+PD-1 inhibitor	II	2017	Active	With or without chemo
**Therapeutic strategy: Tumor vaccine**							
NCT03190265	CY/GVAX+Cyclophosphamide	Previously treated advanced PDAC	Allogeneic cells secreting GM-CSF, as well as alkylating agents	II	2017	Active	Addition to Nivolumab+Ipilimumab+CRS-207
NCT03006302	CY/GVAX	Advanced PDAC	Allogeneic cells secreting GM-CSF	II	2016	Active	Addition to Epacadostat+Pembrolizumab+CRS-207 (IDO1 inhibitor+PD-1 inhibitor+Listeria monocytogenes-expressing mesothelin (tumor vaccine)
NCT02705196	LOAd703+chemo	Early-stage resectable PDAC	Recombinant adenovirus (intratumoral injection)	I/II	2016	Active	With or without atezolizumab (PD-L1 inhibitor)
**Therapeutic strategy: Multiple immunomodulatory approaches**							
NCT02583477	MEDI4736+AZD5069	Advanced PDAC	CXCR2 inhibitor+PD1 inhibitor	I/II	2015	Active	Compared to MEDI4736+ nab-paclitaxel+ gemcitabine
NCT02826486	BL-8040+pembrolizumab	Advanced PDAC	CXCR4 inhibitor+PD1 inhibitor	II	2016	Active	Compared to BL-8040+5-FU
NCT02826486	BL-8040+ Pembrolizumab	Advanced PDAC	CXCR4 inhibitor+PD-1 inhibitor	II	2016	Active	With or without chemo (Onivyde)
NCT03376659	CV301+Durvalumab	Advanced PDAC and CRC	PD-L1 inhibitor+recombinant polypeptides	I/II	2017	Active	With or without chemo
NCT04191421	Siltuximab+ Spartalizumab	Advanced PDAC	IL-6 inhibitor+PD-1 inhibitor	I/II	2019	Recruiting	None
NCT05419479	APX005M+Domvanalimab+Imberelimab	Advanced PCa	CD40 agonist+TIGIT inhibitor+PD1 inhibitor	I/II	2022	Recruiting	With or without chemo
**Therapeutic strategy: The combination of stromal depletion and immune modulation**							
NCT04171219	Talabostat/Pembrolizumab	Advanced PDAC	FAP inhibitor combining PD1 inhibitor	II	2019	Recruiting	None
NCT04177810	Plerixafor/Cemiplimab	Advanced PDAC	CXCR4 inhibitor/PD1 inhibitor	II	2019	Recruiting	None

* As of 30 June 2022. Abbreviations: CRC, colorectal cancer; CCR, chemokine receptors; CXCR4, C-X-C chemokine receptor type 4; chemo, chemotherapy; DNMT, DNA methyltransferase; Gem, gemcitabine; GM-CSF, granulocyte-macrophage colony-stimulating factor; IL-6, Interleukin 6; MPS, methoxsalen, phenytoin, and sirolimus; Nab-P, nab-paclitaxel; PD-1, Programmed cell death protein 1; PCa, pancreatic cancer; PDAC, pancreatic ductal adenocarcinoma; SCLC, small cell lung cancer; TIGIT, T cell immunoreceptor with immunoglobulin and immunoreceptor tyrosine-based inhibitory motif (ITIM) domain.

## Data Availability

Not Applicable.

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
