# Peer review of "Management of Advanced Pancreatic Cancer through Stromal Depletion and Immune Modulation"

_medicina, 2022, doi:10.3390/medicina58091298_

Round 1

Reviewer 1 Report

Dear editor,

The manuscript entitled ‘..’ is quite novel. However, there are points required attention.

Abstract line-20. ‘in-terpersonal’ is wrong. The correct is ‘inter-personal’. Line-23. The word ‘

 ‘stock' is not correct. Replacement is required.

Introduction. Line-46. Correction is suggested here. The 5-year overall survival for PDAC is 8%9% based on the recent reference doi:10.1002/jbt.22900.

The section 3.2. is not discussing about ‘Intertumoral characteristics. It is referred to the activity of cells within the TME of pancreatic cancer.

Line 136 ‘Current therapeutic strategies aim to transform the "immune-cold" tumoral niche into an 136 immunogenic one by, for example, breaking down the thick stroma of the ECM’. The correct sentence is ‘pancreatic caner is an example of cancers with cold immunity, represented by limited infiltration of effector T cells as well as low expression of PD-L1, thereby eliciting low responses to immune checkpoint inhibitor therapy. Thus, a suggested strategy is to bring back the normalization in the tumor immune ecosystem and turn the immune backbone of the tumor into hot, so it would be more responsive to therapy (DOI: 10.1002/jbt.22708).

Page 4, line-153. ‘Pancreatic cancer tumors’ is wrong. Simply say ‘pancreatic cancer’.

Page-4, line-177. ‘This result was not completely unexpected’ what does that mean?

Table-1: The numbering, description and abbreviation of the table must be a part of that not a separate note. In this table, the reader is looking for the outcomes of clinical trials. Where is the outcomes?

Page-6: The authors here are discussing about targeting desmoplasia through addressing the key producers, namely CAFs. Recently, a novel approach to target the fibrotic stroma is to use ‘stroma normalization strategies’ rather than ‘disrupting the dense stroma’ The authors are suggested to discuss about this approach, as it is also important for other events.

Page-7, line-243. ‘Nevertheless, preclinical and clinical trials continue in PDAC animal models’. Please correct as ‘are continued’

Page-8. Immunotherapeutic approaches. It is suggested to put the immune checkpoint blockade (ICB), Cancer vaccine and Cytokines and chemokine based therapy in separate sub-headings

Sincerely

Author Response

  1. Abstract line-20. ‘in-terpersonal’ is wrong. The correct is ‘inter-personal’. Line-23. The word ‘

Reply:Thanks, corrected.

  1. ‘stock' is not correct. Replacement is required.

Reply:Thanks,corrected.

  1. Line-46. Correction is suggested here. The 5-year overall survival for PDAC is ∼8%–9% based on the recent reference doi:10.1002/jbt.22900.

Reply:Thanks for bringing this up. In the first paragraph of Introduction, “Despite the robust growth in basic research on pancreatic cancer and improvements in surgical techniques, the 5-year survival rate for patients remains relatively low, ap-proximately 8 to 9 per 100,000, lower than the survival rates for most other can-cers.” has refered to the overall mortality rate of pancreatic cancer. Here “Even for the small subset of patients who can undergo surgical resection, the 5-year survival rate remains lower than 20%” was referring to the mortality rate of resectable early-stage cancer.

  1. The section 3.2. is not discussing about ‘Intertumoral characteristics. It is referred to the activity of cells within the TME of pancreatic cancer.

Reply:Thanks. This part has been reorganized and renamed with newly added content.

  1. Line 136 ‘Current therapeutic strategies aim to transform the "immune-cold" tumoral niche into an 136 immunogenic one by, for example, breaking down the thick stroma of the ECM’. The correct sentence is ‘pancreatic caner is an example of cancers with cold immunity, represented by limited infiltration of effector T cells as well as low expression of PD-L1, thereby eliciting low responses to immune checkpoint inhibitor therapy. Thus, a suggested strategy is to bring back the normalization in the tumor immune ecosystem and turn the immune backbone of the tumor into hot, so it would be more responsive to therapy (DOI: 10.1002/jbt.22708).

Reply:Thanks. This is a comprehensive and important review. We have corrected the sentence here and cited this paper.

  1. Page 4, line-153. ‘Pancreatic cancer tumors’ is wrong. Simply say ‘pancreatic cancer’.

Reply:Thanks, corrected.

  1. Page-4, line-177. ‘This result was not completely unexpected’ what does that mean?

Reply:We were trying to see the failure of Shh inhibition was not a successful anti-cancer strategy was reasonable and foreseeable. We have re-phrased this sentence.

  1. Table-1: The numbering, description and abbreviation of the table must be a part of that not a separate note. In this table, the reader is looking for the outcomes of clinical trials. Where is the outcomes?

Reply:Thanks, this is a critical comment. We did not add the table to results of some clinical trials. It was a summary of ongoing clinical trails that were related to the topics to display and the current research hotspots so that people can keep track of the in the future. The clinical trials that have been completed with results were refered to in the main body as references.

  1. Page-6: The authors here are discussing about targeting desmoplasia through addressing the key producers, namely CAFs. Recently, a novel approach to target the fibrotic stroma is to use ‘stroma normalization strategies’ rather than ‘disrupting the dense stroma’ The authors are suggested to discuss about this approach, as it is also important for other events.

Reply: Thanks. This is a critical concept and we have rearranged this section with newly added literatures.

  1. Page-7, line-243. ‘Nevertheless, preclinical and clinical trials continue in PDAC animal models’. Please correct as ‘are continued’

Reply: Thanks. Corrected.

  1. Page-8. Immunotherapeutic approaches. It is suggested to put the immune checkpoint blockade (ICB), Cancer vaccine and Cytokines and chemokine based therapy in separate sub-headings

Reply: Thanks, this section has been re-organized and renamed.

Reviewer 2 Report

In this manuscript, the authors summarized the historical and current treatments for and subtyping of pancreatic cancer, and understanding of the tumor microenvironment (TME). Furthermore, the authors introduced several new drug targets to treat pancreatic cancer, with a particular focus on drugs targeting TME and immunotherapeutic pathways. In general, this review manuscript is well organized and well written to cover this topic. Reference papers are appropriately cited to reflect the current research progress. However, there are some parts that could be elaborated and/or added to make this manuscript more comprehensive.

1.     In section 3.2, the authors tried to discuss the TME components that may influence pancreatic cancer biology. In the first paragraph, Cancer-associated fibroblasts (CAFs) are discussed in detail. However, the discussion of other TME component cells, such as tumor-associated macrophages, dendritic cells, natural killer cells, T and B cells, etc., is not enough. Each cell type has its unique features in pancreatic cancer TME and contributes collectively to the immunosuppressive setting. Therefore, this part should be elaborated with more reference papers cited.

2.     In section 5.2, when talking about immunotherapeutic approaches, the cGAS-STING DNA-sensing pathway and anti-tumor immunity are emerging hot topics in cancer immunotherapy. STING agonists are also under development as single or combined therapeutic drugs. The authors should add some content to discuss this field to make the manuscript more inclusive.

3.     Minor language editing is necessary to correct some typos and syntactic errors.

Author Response

  1. In section 3.2, the authors tried to discuss the TME components that may influence pancreatic cancer biology. In the first paragraph, Cancer-associated fibroblasts (CAFs) are discussed in detail. However, the discussion of other TME component cells, such as tumor-associated macrophages, dendritic cells, natural killer cells, T and B cells, etc., is not enough. Each cell type has its unique features in pancreatic cancer TME and contributes collectively to the immunosuppressive setting. Therefore, this part should be elaborated with more reference papers cited.

Reply: Thanks. This section has been reorganized with newly added content referred to in the comment. We are looking forward to further comments.

  1. In section 5.2, when talking about immunotherapeutic approaches, the cGAS-STING DNA-sensing pathway and anti-tumor immunity are emerging hot topics in cancer immunotherapy. STING agonists are also under development as single or combined therapeutic drugs. The authors should add some content to discuss this field to make the manuscript more inclusive.

Reply: This is a hot topic in this area. We added a new section to elaborate on this at the end of this part.

  1. Minor language editing is necessary to correct some typos and syntactic errors.

Reply: Thanks. We have corrected the typos and syntactic errors as much as possible and will continue to working hard on this.

Round 2

Reviewer 1 Report

accept